# The Effect of Mammalian Sex Hormones on Polymorphism and Genomic Instability in the Common Bean (*Phaseolus vulgaris* L.)

**DOI:** 10.3390/plants11152071

**Published:** 2022-08-08

**Authors:** Aras Türkoğlu, Kamil Haliloğlu, Özge Balpinar, Halil Ibrahim Öztürk, Güller Özkan, Peter Poczai

**Affiliations:** 1Department of Field Crops, Faculty of Agriculture, Necmettin Erbakan University, 42310 Konya, Türkiye; 2Department of Field Crops, Faculty of Agriculture, Ataturk University, 25240 Erzurum, Türkiye; 3Department of Biology, Faculty of Science, Cankiri Karatekin University, 18200 Çankırı, Türkiye; 4Hemp Research Institute, Ondokuz Mayıs University, 55200 Samsun, Türkiye; 5Health Services Vocational School, Erzincan Binali Yıldırım University, 24100 Erzincan, Türkiye; 6Department of Biology, Faculty of Science, Ankara University, 06100 Ankara, Türkiye; 7Botany Unit, Finnish Museum of Natural History, University of Helsinki, P.O. Box 7, FI-00014 Helsinki, Finland; 8Institute of Advanced Studies Kőszeg (iASK), P.O. Box 4, H-9731 Kőszeg, Hungary

**Keywords:** CRED-iPBS, epigenetics, genotoxic, in vitro

## Abstract

Mammalian sex hormones are steroid-structured compounds that support the growth and development of plants at low concentrations. Since they affect the physiological processes in plants, it has been thought that mammalian sex hormones may cause modifications to plant genomes and epigenetics. This study aims to determine whether different mammalian sex hormones (17 β-estradiol, estrogen, progesterone, and testosterone) in several concentrations (0, 10^−4^, 10^−6^, and 10^−8^ mM) affect genetic or epigenetic levels in bean plants, using in vitro tissue cultures from plumule explants. We investigated levels of DNA damage, changes in DNA methylation and DNA stability in common bean exposed to mammalian sex hormones (MSH) using inter-primer binding site (iPBS) and Coupled Restriction Enzyme Digestion-iPBS (CRED-iPBS) assays, respectively. The highest rate of polymorphism in iPBS profiles was observed when 10^−4^ mM of estrogen (52.2%) hormone was administered. This finding indicates that genetic stability is reduced. In the CRED-iPBS profile, which reveals the methylation level associated with the DNA cytosine nucleotide, 10^−4^ mM of estrogen hormone exhibited the highest hypermethylation value. Polymorphism was observed in all hormone administrations compared to the control (without hormone), and it was determined that genomic stability was decreased at high concentrations. Taken together, the results indicate that 17 β-estradiol, estrogen, progesterone, and testosterone in bean plants affect genomic instability and cause epigenetic modifications, which is an important control mechanism in gene expression.

## 1. Introduction

Hormones are among the most important physiological factors that play a role in plant growth and development processes [1]. Phytohormones such as auxin, cytokinin, and gibberellin are essential in many growth and development periods, for example, from germination to rooting, and from shoot formation to flowering [2]. In addition to the naturally occurring phytohormones in plants, mammalian sex hormones are also important. These structures, expressed as phytosterols, are thought to have roles related to defense and signal transmission in plants [3]. Mammalian sex hormones, which are detected in most plants, affect growth as well as development and flowering processes, depending on their concentrations [4,5]. Mammalian sex hormones are released into the environment from natural and anthropogenic sources. In addition, these hormones can be treated as external application to plants [6]. The prior studies have been conducted on the effects of MSHs on physiological and genetic mechanisms in plants [7,8,9]. The effects of mammalian sex hormones can occur on DNA at the molecular level. Epigenetic changes are effective in regulating gene expression without changes in the DNA sequence. Such changes can be described as modifications such as histone modifications (acetylation, methylation, phosphorylation, and ubiquitination reactions), remodeling of chromatin, and methylation of DNA [10]. Epigenetic modifications are essential for plants to adapt to their environmental conditions [11]. It has been shown that plants can maintain their survival circulation with epigenetic modifications in response to environmental conditions [12]. Notably, there is evidence of strong interactions between plant hormones and epigenetic signals [13]. Hence, it was hypothesized that the mammalian sex hormones administered in this study might also cause epigenetic modifications.

Plant tissue culture is a commercially important micro-propagation technique that allows plants to be produced throughout the year regardless of external conditions [14,15]. In vitro culture and regeneration of plant cells is a method of asexual reproduction involving the mitotic division of cells whose purpose is the clonal reproduction of genetically uniform plants. This goal is the foundation of the micropropagation industry and provides the technical basis for genetic manipulation. However, uncontrolled variations (such as chromosomal rearrangements, loss or duplication of DNA fragments, minor mutations, and somaclonal variation) can occur during the growth conditions and material management stage of plants during the in vitro stages. In other words, tissue culture regenerators are not always genotypically and phenotypically similar [16,17,18]. Recent studies have shown that phenotypic and genotypic variations that occur in plants during in vitro production also cause changes in the biochemistry [19] and metabolites of plants. In general, it has been argued that metabolites are more closely related to phenotypes than genes, and that metabolomics is the link between genotype and phenotypes [20]. Due to changes in the tissue culture micro-environment, plant cells are subjected to additional stress that causes genetic and epigenetic imbalances in regenerants. These changes lead to tissue-culture-induced variations, also known as somaclonal variations, to categorically indicate the inducing environment [21]. Tissue-culture-induced variations include molecular and phenotypic changes induced in vitro due to continuous subculturing and stress derived from tissue cultures, which can induce epigenetic variations such as altered DNA methylation patterns [20]. It has been reported that DNA methylation is the most common covalent base modification found in various taxa or species [22]. Since DNA methylation plays an important role in gene expression and regulation of plant development, variants that arise during the tissue culture process due to inherited methylation changes may contribute to intra-specific phenotypic variation [23]. Here, we review the aspects of tissue-culture-induced variation in relation to DNA methylation and its impact on crop improvement programs [22].

DNA methylation includes methylation in DNA at cytosine residues [24]. Cytosine methylation is a flexible epigenetic regulatory mechanism that controls gene expression by inhibiting the binding of proteins to DNA and altering the structure of associated chromatin. In plants, DNA methylation can occur on cytosines in any context, and CG is the most commonly methylated dinucleotide [25,26]. CG and non-CG methylation can silence transposons and pseudogenes, and can regulate plant development and tissue-specific gene expression [27]. Current evidence indicates that the altered pattern of cytosine methylation is much more frequent in the plant genome in vitro, resulting in different phenotypic changes [28]. Tissue culture-induced mutations such as activation of transposable elements, chromosome breakage, and changes in DNA sequence are hypothesized to arise as a result of DNA methylation, which ultimately leads to a high rate of phenotypic variation [29]. DNA methylation is the most common covalent base modification in various taxa [5]. Specific DNA methylases produce several methylated bases during the post-replication state of DNA modification, with 5 methyl-cytosine (5-mC) representing the most common form in higher plants and mammals [5].

Various methods are available to detect changes in the genome-wide methylation pattern of plants in tissue culture media, in light of extensive knowledge of the genome sequence of most organisms [30]. However, these include a modified amplified fragment length polymorphism (AFLP), which does not require genome sequencing, and the methylation-sensitive amplification polymorphism (MSAP) technique [31]. In addition, techniques such as high performance capillary electrophoresis (HPCE) [32] and high performance liquid chromatography (HPLC) [33] are successfully used to detect cytosine methylation. Coupled restriction enzyme digestion and inter-primer binding site (iPBS) (CRED-iPBS) is another significant technique for studying the methylation status in plants. iPBS-retrotransposons are a PCR-based marker system based on the presence of tRNA as a reverse transcriptase primer binding. It is stated that iPBS markers are highly effective and reliable for detection of polymorphism and clonal differentiation resulting from various retrotransposon activities and the retrotransposon recombination site [34]. The CRED-iPBS technique also has been successfully applied in many studies of different plant species to determine their methylation patterns. The CRED-iPBS technique is used to detect methylation changes between different tissues or different stages of development after restriction digestion of DNA with methylation sensitive enzymes such as *HpaII* and *MspI* [35,36,37].

The primary objective of this study was to elucidate the effects of mammalian sex hormones (MSHs) on plant tissue culture, genomic instability, and DNA methylation of bean plants. Specifically, epigenetic modifications in the genomic DNA of bean plumule explants exposed to different concentrations of various mammalian sex hormones were investigated. The methylation states of the bean explant DNA were determined using the CRED-iPBS technique. This study is the first to examine the epigenetic changes caused by MSHs in the bean genome with the iPBS and CRED-iPBS methods.

## 2. Results

### 2.1. iPBS Analysis

This study investigated the genetic and epigenetic effects of different concentrations of mammalian sex hormones on bean plants using iPBS and CRED-iPBS methods. Significant changes in the bean plant iPBS profile were detected in all applied MSHs. These variations, seen as both the disappearance of standard bands and the emergence of new bands, were calculated as the genomic template stability (GTS) value compared to the control (MS without any hormone) group (Table 1, Figure 1). While the control had a total of 69 bands in the iPBS profile, this number varied between 18–20, 21–26, 19–30, and 24–36 in 17β-estradiol, progesterone, testosterone, and estrogen hormones, respectively, depending on the concentrations. Polymorphic bands showed increased diversity in the bean genome after administering mammalian sex hormones. Depending on the concentrations and MSH type, the polymorphism rates of the bean plant also changed. In this case, the highest polymorphism was observed at a concentration of 10^−4^ mM of estrogen hormone (52.2% polymorphism rate), followed by testosterone hormone at a concentration of 10^−4^ mM (43.5% polymorphism rate) (Table 1). The percentage of GTS was also calculated when detecting changes in the iPBS profile. The percent GTS is a value that allows qualitative quantification of changes in the iPBS profile. GTS calculation was performed for all 10 iPBS primers used. There was a negative correlation between the percentage of GTS and MSH concentrations. The highest GTS value was detected at 10^−8^ and 10^−6^ mM (73.9% GTS rate) of 17β-estradiol, while the lowest was found at a concentration of 10^−8^ mM (47.8% GTS rate) of estrogen. The results indicate that under tissue culture conditions, the iPBS profile of all MSHs administered to the bean plant showed not only significant changes, but also polymorphic bands in the bean genome after application of mammalian sex hormones.

### 2.2. CRED-iPBS Analysis

CRED-iPBS analysis was used to examine the effect of mammalian sex hormones on methylation rates in the bean plant. The results show that DNA hypermethylation/hypomethylation was dependent on MSH type and concentration compared to the PCR production obtained from the control DNA. Results of the CRED-iPBS analysis are expressed as polymorphism percentage in *MspI* and *HpaII* digested CRED-iPBS assays (Table 2, Figure 2). The rate of polymorphism associated with the MspI enzyme was found to be higher compared to *HpaII.* DNA hypermethylation was mostly observed at 10^−4^ mM of mammalian sex hormones. These hypermethylation values were 39.3%, 47.5%, 47.5%, and 52.5% for 17β-estradiol, progesterone, testosterone, and estrogen hormones, respectively. DNA hypomethylation was detected at the lowest concentration of mammalian sex hormones, 10^−8^ mM. Hypomethylation values were determined as 23.7%, 27.9%, 35.6%, and 29.51% for 17β-estradiol, progesterone, testosterone, and estrogen hormones, respectively (Table 2). A clear decrease was seen in average polymorphism percentage and methylation status. Thus, it can be concluded that MSH at low concentrations had a protective role in hypermethylation. The polymorphism percentage gradually decreased at low MSH concentrations (Table 2). These results show that at low concentrations, MSHs have an antagonistic effect against epigenetic and genotoxic effects.

## 3. Discussion

Depending on the concentrations and MSH type, the polymorphism rates of the bean plant also changed. In this case, the highest polymorphism was observed at a concentration of 10^−4^ mM of estrogen hormone. The highest GTS value was detected at concentrations of 10^−8^ and 10^−6^ mM of 17β-estradiol. Under the tissue culture system, explants undergo either direct or indirect organogenesis and somatic embryogenesis [38]. The proliferating cells in the callus undergo redifferentiation, which leads to organogenesis or plantlet regeneration by the application of plant growth regulators (PGRs) in culture [39]. These changes can also occur when MSH has been applied. The process of differentiation and redifferentiation under artificial conditions during in vitro culture exerts traumatic stress on plant cells, initiating mitotic and meiotic inherited genetic and epigenetic variations [29]. Moreover, tissue-culture-induced variations can also occur due to epigenetic regulation, which can cause the level of DNA methylation to change permanently [40]. Mammalian sex hormones stimulate the growth and development of plants, especially in low concentrations. At the same time, these hormones play a role in many physiological processes of plants, such as germination and flowering [41]. Plants are affected by environmental factors. As a result, epigenetic changes such as DNA methylation, small RNAs, and histone modifications occur in plant genomes [42].

The administered MSH constructs caused changes in both the bean plant’s genetic structure and epigenetic profile. It is thought that this may be because MSHs affect plants’ inorganic contents. For example, a study conducted on bean plants determined that mammalian sex hormones cause changes in the ratio of inorganic substances in the plant content [43]. Inorganic substances are essential structures that affect the functioning of enzymes. Plants need inorganic elements (such as P, S, K, Fe, and Ni) for synthesis reactions such as photosynthesis, protein synthesis, and nucleic acid synthesis. It has been reported that there is an important relationship between inorganic elements and the speed of metabolic pathways in all living things. However, inorganic elements have a vital role in the formation of organic substances. For example, the inorganic element P is a vital component of DNA and RNA [44]. For this reason, MSHs can affect every physiological process where enzymes have an effect. The stresses that occur during plantlet regeneration through the tissue culture process affect the normal functioning of cell organelles, but first the plasma membrane and cell wall sense the stress and produce reactive oxygen species (ROS) [45]. In another study, it was reported that some mammalian sex hormones (MSHs) were successful in preventing genetic and epigenetic changes caused by certain chemicals in plants [46]. In addition, it is reported that MSHs stimulate antioxidant defense systems in plants [47]. In addition, it is known that under stress conditions, hypomethylation and hypermethylation of DNA and gene expression can change in plants [48].

The CRED-iPBS technique, which is associated with DNA cytosine methylation, detects changes and variations in the sample DNA. A hypermethylation state means that a gene is silenced and hence not being expressed. In hypomethylation, an increase in the gene’s activity is observed [49]. In this study, hypermethylation occurred at high MSH concentrations. This is probably due to the frequent use of MSH in callus and cell cultures, which induces genetic abnormalities such as polyploidy and DNA endo-replication. These findings are similar to those reported in the literature, from studies that determined epigenetic modifications in different plants under abiotic stress conditions [35,36,37,50,51].

## 4. Materials and Methods

### 4.1. Plant Material and Culture Conditions

The Elkoca cultivar of the bean plant was used to evaluate the effect of mammalian sex hormones. Plant material was obtained from Atatürk University Agricultural Engineering Department of Field Crops. Seeds for germination were prepared in the dark, on MS [25] free medium containing 20 g L^−^^1^ sucrose, 2 g L^−^^1^ phytagel, 1.95 g L^−^^1^ MES [(2-(N-morpholino) ethane sulfonic acid)]. The pH of the nutrient medium was adjusted to 5.7–5.8. Media solutions containing basal salts and solidifying agent were autoclaved at 121 °C for 20 min for sterilization. Plumule explants that were isolated from the germinated seeds at the end of the fourth day of culturing in hormone-free MS medium were incubated at 25 ± 1 °C for 4 days. After four days of culture, 10 prepared explants (each with four replications) were placed into petri dishes containing MS medium supplemented with 20 g L^−^^1^ sucrose, 2 g L^−^^1^ phytagel, 1.95 g L^−^^1^ MES [(2-(N-morpholino) ethane sulfonic acid)], and different concentrations (control (without hormone), 10^−^^4^, 10^−^^6^, and 10^−^^8^ mM) of one of four mammalian sex hormones [estrogen (C_18_H_22_O_2_: 270.37 g/mol; Sigma Aldrich; Product Number: E9750), progesterone (C_21_H_30_O_2_: 314.46 g/mol; Sigma Aldrich; Product Number: Y0001665), 17 β-estradiol (C_18_H_24_O_2_: 272.38 g/mol; Sigma Aldrich; Product Number: E2758), and testosterone (C_19_H_28_O_2_: 288.42 g/mol; Sigma Aldrich; Product Number: Y0002163)]. All samples, including the control group, originated from the same donor plant. Media solidification, pH adjustment, and sterilization were carried out as described for callus induction media. Regeneration callus cultures were incubated in a growth chamber at 25 ± 1 °C with a photoperiod of 16 h light (62 μmol m^−2^ s^−1^) and 8 h dark.

### 4.2. Genomic DNA Isolation

Genomic DNA was isolated from the plumule explants treated for four weeks with mammalian sex hormones using the method explained by Zeinalzadehtabrizi et al. [52], with minor modifications. Subsequently, the DNA was stored at −20 °C for further use. The amount of DNA was determined with the using of NanoDrop (Qiagen Qiaxpert) device and the quality of DNA was tested using 1.5% agarose gel electrophoresis [53]

### 4.3. iPBS-PCR Amplification

Twenty primers were tested for iPBS-PCR amplification; only 10 primers produced clear and polymorphic banding patterns in all treatments (Table 3). The PCR master mix consisted of 10× buffer, 2 mM MgCl2, 0.25 mM of each dNTPs, 2 μM (20 pmol) primer, 0.5 U Taq polymerase, and 1 μL of 50 ng/μL template DNA. The following amplification conditions were used: initial denaturation at 95 °C for 3 min, 38 cycles of 15 s at 95 °C, 60 s at 51–56 °C, and 60 s at 72 °C, followed by 5 min at 72 °C. CRED-iPBS PCR samples were visualized on a 3% agarose gel [37].

### 4.4. CRED-iPBS Amplification

A total of 1000 ng of DNA was cut using 1 U of HpaII or MspI enzymes. This process aimed to obtain template DNA according to the manufacturer’s (Thermo Scientific) instructions. Subsequently, the fragmented DNA was amplified using the 10 iPBS primers listed above. The PCR master mix consisted of 10× buffer, 2 mM MgCl2, 0.25 mM of each dNTPs, 2 μM (20 pmol) primer, 0.5 U Taq polymerase, and 1 μL of 50 ng/μL template DNA. The following amplification conditions were used: initial denaturation at 95 °C for 3 min, 38 cycles of 15 s at 95 °C, 60 s at 51–56 °C, and 60 s at 72 °C, followed by 5 min at 72 °C. CRED-iPBS PCR samples were visualized on a 3% agarose gel [35].

### 4.5. CRED-iPBS Analysis

Both iPBS and CRED-iPBS bands were analyzed using TotalLab TL120 software (Nonlinear Dynamics Ltd.R). Genomic template stability (GTS %) was calculated according to band profiles using the following formula: GTS = (1 − a/n) × 100, where “a” corresponds to the mean number of polymorphic bands, and “n” represents the total number of bands in the control. Polymorphisms in the iPBS profile are detected compared to the control, either as a new band not present in the control, or as the absence of a band that is present in the control. Means were calculated for each experimental group, and changes in the means of each group were calculated as a percentage compared to the control. Mean polymorphism values for the CRED-iPBS analysis were calculated using the formula 100 × a/n [36].

## 5. Conclusions

Recently, DNA methylation has been recognized as the main regulatory epigenetic mechanism associated with various regulatory gene functions during the tissue culture process. Although many studies have been undertaken on tissue-culture-induced variations, including DNA methylation, the process is still far from fully understood. Various molecular approaches have been used to confirm genetic fitness in tissue culture plants. Epigenetic factors have also been found to be associated with phenotypic variation. In this study, the effects of four different MSHs (17β-estradiol, progesterone, testosterone, and estrogen) at four different concentrations (0, 10^−^^4^, 10^−^^6^, and 10^−^^8^ mM) on the genetic and epigenetic stability of bean plants were investigated by iPBS and CRED-iPBS analysis. This is the first study to use both iPBS and CRED-iPBS methods for detecting DNA alteration in beans under MSH type and concentration in in vitro condition. The findings and genetic variations obtained from MSH applications can be used to induce the adaptation process and development of the bean plant. In conclusion, this study reveals that MSHs play active roles in genomic stability and cause genetic/epigenetic modifications.

## Figures and Tables

**Figure 1 plants-11-02071-f001:**
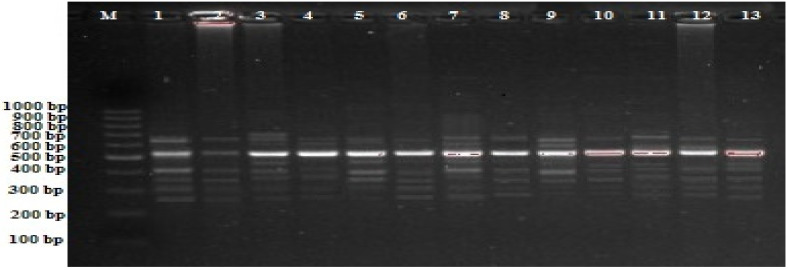
iPBS profiles for various experimental groups with 2075 primers in bean M; 100–1000 bp DNA ladder, 1; control (MS without any hormone) 2; MS medium containing 10^−^^8^ mM 17-β-estradiol, 3; MS medium containing 10^−^^6^ mM 17-β-estradiol, 4; MS medium containing 10^−^^4^ mM 17-β-estradiol, 5; MS medium containing 10^−^^8^ mM progesterone, 6; MS medium containing 10^−^^6^ mM progesterone, 7; MS medium containing 10 ^4^ mM progesterone, 8; MS medium containing 10^−^^8^ mM testosterone, 9; MS medium containing 10^−^^6^ mM testosterone, 10; MS medium containing 10^−^^4^ mM testosterone, 11; MS medium containing 10^−^^8^ mM estrogen, 12; MS medium containing 10^−^^6^ mM estrogen, 13; MS medium containing 10^−^^4^ mM estrogen.

**Figure 2 plants-11-02071-f002:**
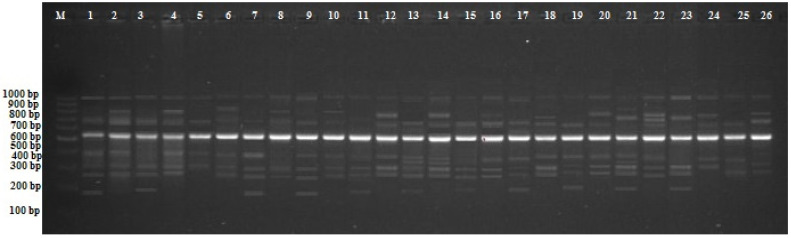
CRED-RA profiles for various experimental groups with OPH-17 primers in bean. M; 100–1000 bp DNA ladder,1; control (MS without any hormone) Msp I 2; control (MS without any hormone) Hpa II, 3; MS medium supplemented with 10^−^^8^ mM 17-β-estradiol Msp I, 4; MS medium supplemented with 10^−^^8^ mM 17-β-estradiol Hpa II, 5; MS medium supplemented with 10^−^^6^ mM 17-β-estradiol Msp I, 6; MS medium supplemented with 10^−^^6^ mM 17-β-estradiol Hpa II, 7; MS medium supplemented with 10^−^^4^ mM 17-β-estradiol Msp I, 8; MS medium supplemented with 10^−^^4^ mM 17-β-estradiol Hpa II, 9; MS medium supplemented with 10^−^^8^ mM progesterone Msp I, 10; MS medium supplemented with 10^−^^8^ mM progesterone Hpa II, 11; MS medium supplemented with 10^−^^6^ mM progesterone Msp I, 12; MS medium supplemented with 10^−^^6^ mM progesterone Hpa II, 13; MS medium supplemented with 10^−^^4^ mM progesterone Msp I, 14; MS medium supplemented with 10^−^^4^ mM progesterone Hpa II, 15; MS medium supplemented with 10^−^^8^ mM testosterone Msp I, 16; MS medium supplemented with 10^−^^8^ mM testosterone Hpa II, 17; MS medium supplemented with 10^−^^6^ mM testosterone Msp I, 18; MS medium supplemented with 10^−^^6^ mM testosterone Hpa II, 19; MS medium supplemented with 10^−^^4^ mM testosterone Msp I, 20; MS medium supplemented with 10^−^^4^ mM testosterone Hpa II, 21; MS medium supplemented with 10^−^^8^ mM estrogen Msp I, 22; MS medium supplemented with 10^−^^8^ mM estrogen Hpa II, 23; MS medium supplemented with 10^−^^6^ mM estrogen Msp I, 24; MS medium supplemented with 10^−^^6^ mM estrogen Msp I, 25; MS medium supplemented with 10^−^^4^ mM estrogen Hpa II, 26; MS medium supplemented with 10^−^^4^ mM estrogen Msp I.

**Table 1 plants-11-02071-t001:** Molecular sizes (bp) of present/absent bands in iPBS profiles after application of different MSHs and different concentrations in treated *Phaseolus vulgaris* L. under tissue culture conditions.

iPBS Primer	^*^+/- ^**^	Control ^***^	Experimental Groups											
			17 β-Estradiol			Progesterone			Testosterone			Estrogen		
			10^−^^8^ mM	10^−6^ mM	10^−^^4^ mM	10^−^^8^ mM	10^−6^ mM	10^−^^4^ mM	10^−^^8^ mM	10^−6^ mM	10^−^^4^ mM	10^−^^8^ mM	10^−6^ mM	10^−^^4^ mM
2075	+	6	-	689; 590	590	590	-	590	-	571	461	454	-	447
	-		-	-	-	-	-	-	-	-	-	-	-	-
2077	+	6	-	379	920; 918; 659; 467; 372	940; 843; 379	950; 925; 857; 475; 379	950; 875; 900; 475	-	491; 379	960; 850	491	960; 880; 483; 386	900; 850
	-		813; 700;	-	-	-	-	-	-	-	311	-	-	-
2087	+	4	-	-	900; 452; 416	-	425	900; 480	-	-	940	960; 320	980; 960;	1320
	-		-	-	-	831; 500	-	727; 648	831; 500	831; 727; 500	727; 648; 500	-	727	831; 727
2278	+	9	-	950	900; 950	1357	920; 800; 750	1357	920; 800	920	920; 800	920; 800	920; 900	920; 800; 750
	-		752; 520	520	-	752	-	520	-	520	752; 400; 310	520; 382	-	752; 664; 400; 310
2375	+	10	-	1371	1371; 970; 852; 536	1357	975; 885; 800; 757; 725; 575	957	975; 985; 857	971	975; 885	957; 900; 814	985; 885	985; 914; 828
	-		779; 607	607	-	779	-	607	-	607	779; 718; 486; 308	607; 382	-	779; 718; 664; 422; 382
2377	+	9	-	-	-	-	-	-	-	-	985	-	-	-
	-		957; 970; 710; 582; 431	970; 582; 487; 431	431	900; 710; 637; 582; 520; 487; 431	487; 431	582; 487; 431	710; 487; 431	487; 431	487; 431	431	582; 487; 431	970; 582
2380	+	5	-	548	-	-	1060; 920	1060; 837	858; 560	940; 858	-	879; 783	980; 950; 900	940; 837
	-		-	-	-	-	-	-	-	-	-	-	-	-
2381	+	9	-	-	-	-	-	-	-	-	-	-	-	-
	-		900; 850; 833; 420; 376	900; 850; 972; 833	472; 376	972; 376	472; 376	950; 900; 872; 472; 420; 324	900; 833; 472; 376	900; 850; 822; 800; 651; 472; 420; 376	900; 850; 833; 472; 376	900; 651; 420; 376; 324	972; 833; 651; 472; 376; 324	900; 822; 783; 651; 472; 420; 376; 324
2382	+	6	936	-	917	-	1040; 413	1000	-	-	-	-	-	917; 809
	-		-	-	-	521	-	521	521	589; 500	870; 589; 521	521	-	-
2384	+	5	-	525; 239	-	-	-	-	-	-	-	759; 553	711; 543; 361	-
	-		282	-	800	282	-	-	392; 282	-	-	-	-	800
Total band		69	18	18	20	21	23	26	19	24	30	24	26	36
Polymorphism (%)			26.1	26.1	29	30.4	33.3	37.7	27.5	34.8	43.5	34.8	37.7	52.2
GTS value (%)			73.9	73.9	71	69.6	66.7	62.9	72.5	65.2	56.5	65.2	62.9	47.8

^*, **^ and ^***^; appearance of a new band, disappearance of a normal band and without hormone, respectively.

**Table 2 plants-11-02071-t002:** Results of CRED-iPBS analysis; molecular size of bands and polymorphism percentage.

iPBS Primer	M^*^/H ^**^	+^***^/- ^****^	Control ^*****^	Experimental Groups											
				17 β-Estradiol			Progesterone			Testosterone			Estrogen		
				10^−8^ mM	10^−6^ mM	10^−4^ mM	10^−8^ mM	10^−6^ mM	10^−4^ mM	10^−8^ mM	10^−6^ mM	10^−4^ mM	10^−8^ mM	10^−6^ mM	10^−4^ mM
2075	M	+	7	-	-	-	-	-	347	-	-	-	626	-	-
		-		-	-	393	-	-	-	-	-	-	-	271	393
	H	+	7	-	-	-	-	-	376	-	-	-	315	-	-
		-		400	393; 271	-	-	-	-	400	-	-	-	400	266; 180
2077	M	+	6	916; 883; 750	816; 700; 618	465	916; 812	933; 866; 850; 716	950; 866;800; 750; 718	916; 900; 918	950; 816; 733; 718; 465	-	933; 900; 833;	933; 800	916; 860; 800; 613;491; 377
		-		356; 318	318	-	318	-	-	415; 312	-	356; 318	-	312	-
	H	+	6	818	1413; 1016	933; 850; 718	916; 916; 838	933;816	933; 833; 718	933; 1050	950; 923; 936	950; 900; 800	916; 816; 780	933; 850; 884; 800; 780; 739	933; 866; 716
		-		-	415; 312	575; 415; 356; 312	-	312	600	318	-	-	415; 312	-	415; 318
2087	M	+	7	-	-	522; 400; 350	-	1675	-	-	-	-	-	-	-
		-		922; 815; 700; 567; 500	1125; 922; 815; 629; 567; 418	1300; 1075	975; 922; 815; 700; 600; 489	922; 815; 629; 567; 500	975; 922; 815; 529; 567	950; 875; 822; 700; 600;	900; 875; 815	925; 822; 815; 700; 629; 500	925; 922; 815;700; 629; 418	900; 875; 822; 815; 700; 600	975; 822; 715; 700; 600
	H	+	8	900; 427	-	-	320	-	-	-	950	320	-	-	-
		-		-	975; 922; 815; 600;	975; 922; 815; 700; 489	900; 815	900; 875; 815	925; 629; 418	925; 822; 815; 629; 567; 418	700; 567; 418	900; 805; 722; 615	900; 875; 822; 700; 600	815; 567; 500; 418	925; 822; 815; 629; 567; 418
2278	M	+	8	-	-	500	-	-	1475	-	-	-	-	-	-
		-		925; 760	825;465	900; 900	900; 750; 650; 450	955; 900; 465	925; 900; 760	900; 800; 750; 450	850; 800	925; 900; 760	-	900; 800;715; 750	900; 815; 750; 450
	H	+	5	950; 427	-	-	310	-	-	-	950	-	-	-	-
		-		-	900; 815	900; 815; 750; 450	900; 815	900; 900; 815	925; 465	760; 465	760; 465	900; 700	950; 900; 750	925; 760; 465	1125; 760
2375	M	+	5	920; 589	920	940	900; 866	1000; 579	900; 820; 747; 589	837	900; 851; 811	980; 920; 661; 589; 454	980; 920; 651; 589	824; 326	851
		-		-	-	-	-	-	-	-	-	-	-	-	-
	H	+	4	-	938; 811	-	837; 700	880; 851; 700	840; 589; 468	920; 454	880; 760; 589	851; 737; 700	-	900; 820; 589	920; 680; 365
		-		866;811; 709	-	405	-	-	-	709	-	-	-	-	-
2377	M	+	6	1262	1287	640; 583; 492; 338; 264	913	887; 1125	1312	926	964; 739; 682; 632	-	-	912	975; 839; 762; 648; 616; 492
		-		431; 394	431; 394	-	-	500; 394	-	-	-	729; 500; 394	-	400	-
	H	+	6	-	987	-	-	739	1262	-	860; 1087; 762	-	1100	975; 887	975
		-		-	548	-	-	-	431; 394	431	-	885; 827; 453; 400	-	-	729; 500
2380	M	+	5	607; 500	1000; 555	-	732; 607; 568; 450	541	889	607	607	-	944; 527	912	944; 880; 613; 580
		-		-	-	-	-	-	-	-	-	846; 400	-	-	-
	H	+	5	-	961	875; 607; 527	600	944	600; 500	-	673	912; 591	-	981	591
		-		-	400	-	-	-	-	400	846	-	-	-	-
2381	M	+	5	-	-	-	-	-	900; 724; 550	-	925; 900; 750; 334	-	-	325; 269	-
		-		611	819; 353	-	833; 466	819	-	921; 633; 537	-	811; 611	819; 427; 353	921; 633	466
	H	+	4	-		-	-	-	900; 679	-	925; 800; 679	-	-	-	-
		-		-	537; 466; 409	537	633	921; 633; 537	-	427	-	633; 537; 466; 409	921; 633; 466; 409	-	819; 427; 353
2382	M	+	4	-	-	753; 529	586; 369	-	-	-	916; 848	-	-	-	545
		-		645	-	-	-	753; 645; 537	940; 753	492	-	940; 753; 645	940; 753; 645; 537	-	-
	H	+	8	-	-	408	-	-	-	-	-	-	529	-	-
		-		940	-	-	900	645	753; 645	940; 537	940;	645	-	940; 753	940; 537
2384	M	+	8	-	-	-	-	-	-	-	-	-	700; 480	-	-
		-		958	958; 408	958; 830; 540; 461; 391; 346	958; 830	958	958; 276	958; 461; 346	958; 346	958; 276	-	958; 346; 281	830; 461; 346
	H	+	6	-	-	-	-	-	-	461	-	-	-	-	920; 452
		-		958; 830; 551; 408	-	-	958; 346; 281	958; 540; 461; 346	-	-	958	958; 346; 281	-	958; 830	-
Polymorphism %		M		37.3	37.3	39.3	41	44.1	47.5	39.3	42.6	47.5	42.4	42.6	52.5
		H		23.7	32.8	36.1	27.9	36.1	37.3	35.6	39	44.3	29.51	41	41.2

^*^, ^**^, ^***^, ^****^ and ^*****^; M—Msp I, H—Hpa II, appearance of a new band, disappearance of a normal band and without hormone, respectively.

**Table 3 plants-11-02071-t003:** Sequence information of 10 iPBS primers and their annealing (Ta) temperatures.

Primer Name	Sequence (5′–3′)	Tm (°C)	CG (%)
iPBS-2075	CTCATGATGCCA	42.1	50
iPBS-2077	CTCACGATGCCA	46.1	58.3
iPBS-2087	GCAATGGAACCA	43.5	50
iPBS-2278	GCTCATGATACCA	42.3	46.2
iPBS-2375	TCGCATCAACCA	45.1	50
iPBS-2377	ACGAAGGGACCA	47.2	58.3
iPBS-2380	CAACCTGATCCA	41.4	50
iPBS-2381	GTCCATCTTCCA	40.9	50
iPBS-2382	TGTTGGCTTCCA	44.9	50
iPBS-2384	GTAATGGGTCCA	40.9	50

## Data Availability

Data is contained within the article.

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
