# Peer review of "The Effect of Mammalian Sex Hormones on Polymorphism and Genomic Instability in the Common Bean (Phaseolus vulgaris L.)"

_plants, 2022, doi:10.3390/plants11152071_

Round 1
Reviewer 1 Report
General.
The idea of analyzing the role of hormones on somaclonal variation is an exciting issue. The information that could be gained from such study would of value and should impact important studies in the field. Having this in mind I am sure that the study may have scientific value. However, there are many aspects that are problematic. I am not convinced that quantification of differences between samples is correct. Furthermore, I think that the differences observed may result from sample sizes of control and tested materials rather that from variation related to hormones. Still, I cannot exclude that but Author should consider how to convinced a Reader that their approach is correct.
In my opinion, the ms requires extensive revision before it could be considered for publication in the journal.
Furthermore, see comments below.
Other issues.
Introduction
Authors: However, tissue culture regenerants are not always genotypically and phenotypically similar.
Comment: Please, add references (additional to the ref. 20) and give descriptive examples indicating what kind of genotypic and morphological changes were evaluated. It is also worth of mentioning the variation affecting biochemistry and metabolome.
Authors: Due to changes in the tissue culture micro-environment, plant cells are subjected to additional stress that causes genetic and epigenetic imbalances in regenerants.
Comment: Please, add references and check the text for plagiarisms.
Authors: These changes lead to tissue culture-induced variations, also known as somaclonal variations, to categorically indicate the inducing environment.
Comment: Please, add references and check the text for plagiarisms
Authors: Tissue culture-induced variations include molecular and phenotypic changes induced in vitro due to continuous subculturing and stress derived from tissue cultures, which can induce epigenetic variations such as altered DNA methylation patterns.
Comment: Please, add references and check the text for plagiarisms
Authors: Reportedly, the change in DNA methylation pattern is much more frequent in the plant genome during the tissue culture process.
Comment: This statement is not true. It depends on species. Please, check barley and triticale data on the issue.
Authors: Since DNA methylation plays an important role in gene expression and regulation of plant development, variants that arise during the tissue culture process due to inherited methylation changes may contribute to intra-specific phenotypic variation.
Comment: I understand this is your suggestion. Do you have any data that support the notion?
Authors: There are various molecular techniques for detecting DNA methylation at different stages of in vitro culture.
Comment: Please, note that you describe them shortly later. The phrase is not needed here.
Authors: Various methods are available to detect changes in the genome-wide methylation pattern in tissue culture plants, most of which are based on extensive knowledge of an organism’s genome sequence [22]. However, methylation-sensitive amplification polymorphism (MSAP) – a modified amplified fragment length polymorphism (AFLP) technique does not require genome sequencing [23].
Comment: the text sound highly familiar. Have you checked it for plagiarisms?
Results & Discussion
General. It would be great if you could separate Result and Discussion section into two independent part. In R section or maybe in methods graphical representation of your whole tissue culture and molecular experiment will be included.
Comment: line 122. What does GTS mean?
Comment: line 129. 10-4 mM of estrogen hormone (52.2%). Should it be 10-4? Explain percentages in brackets, please? How did you get the number? Keep this in mind in the next sentences (130, 134, 135), too.
Authors: There was a negative correlation between the percentage of GTS 133 and MSH concentrations.
Comment: It is not apparent how quantification was conducted. Have you normalized your results somehow? Without it, it would be inadequate to make any comparisons (correlations).
Authors: … such as germination and flowering [33].
Comment: Furthermore, under some concentrations hormones may act as antioxidants as they have p bands. Its worth of tracing this aspect in the literature as antioxidative properties may explain the variation you notice. Suppose, ROS may modify methylated C leading to mutations (at least at CHH sites that are hard for reparation during replication). If ROS scavengers are present such option should be (or one may expect, could be) less pronounced.
Authors: The administered MSH constructs caused changes in both the bean plant’s genetic structure and epigenetic profile. It is thought that this may be because MSHs affect plants up to inorganic content.
Comment: Could you be more precise, please? Do you mean metal ion cofactors? If so, please check my suggestion concerning antioxidants.
Author: For example, a study conducted on bean plants determined that mammalian sex hormones cause changes in the ratio of inorganic substances in the plant content [35]. Inorganic substances are essential structures that affect the functioning of enzymes.
Comment: Again, more info is needed on what you mean by INORGANIC SUBSTANCES. Please, be concrete.
Comment: Figure 1. Would you give more details on control, please? How it is related to other samples? Have the other samples and control originated from the same donor plant?
Comment on materials: In your experiment cultivar rather than genetically and epigenetically uniform plant was used. If so, all your differences compared to the control may be due to sampling variation. If I am correct then your conclusions are doubtful. Please, support evidence that your are right.
Alternatively, you may consider population genetic approach. However, for that you should have quit a larch number of individuals for each experiment, check for PIC, I, He etc. Having stable parameters for control (via checking different sample sizes you may have a background for choosing an adequate sample size for your analysis).
Author Response
Responses to Comments of Reviewer 1
General Response:
Dear reviewer; We tried to respond to your valuable suggestions and comments in the best way possible. I hope it was a successful arrangement. The edits you want are highlighted in yellow background color.
Sincerely
Dr. Aras Turkoglu, Dr. Peter Poczai et al.
|
Comments |
Responses: |
|
1. Authors: However, tissue culture regenerants are not always genotypically and phenotypically similar. Comment: Please, add references (additional to the ref. 20) and give descriptive examples indicating what kind of genotypic and morphological changes were evaluated. It is also worth of mentioning the variation affecting biochemistry and metabolome. |
Added new references and edited sentences…… In vitro culture and regeneration of plant cells is a method of asexual reproduction involving the mitotic division of cells whose purpose is the clonal reproduction of genetically uniform plants. This goal is the foundation of the micropropagation industry and provides the technical basis for genetic manipulation. However, uncontrolled variations (such as chromosomal rearrangements, loss or duplication of DNA fragments, minor mutations, and somaclonal variation) can occur during the growth conditions and material management stage of plants during the in-vitro stages. In other words, tissue culture regenerators are not always genotypically and phenotypically similar. [Sahijram et al., 2003; Oh et al., 2007; Bradaï et al., 2016]. Recent studies have shown that phenotypic and genotypic variations that occur in plants during in vitro production also cause changes in the biochemical [Bednarek and Orłow-ska 2021] and metabolites of plants. In general, it has been argued that metabolites are more closely related to phenotypes than genes, and that metabolomics is the link between genotype and phenotypes [Carrera et al., 2021]. |
|
2. Authors: Due to changes in the tissue culture micro-environment, plant cells are subjected to additional stress that causes genetic and epigenetic imbalances in regenerants.
Comment: Please, add references and check the text for plagiarisms. |
3. Added references …….. …………………..Due to changes in the tissue culture micro-environment, plant cells are subjected to additional stress that causes genetic and epigenetic imbalances in regenerants [Smulders and De Klerk 2011]. Smulders MJM, De Klerk GJ: Epigenetics in plant tissue culture. Plant Growth Regul. 2011, 63(2), 137-146. Necessary checks were made in terms of plagiarism with the “ITHENTICATE” programme. |
|
4. Authors: These changes lead to tissue culture-induced variations, also known as somaclonal variations, to categorically indicate the inducing environment.
Comment: Please, add references and check the text for plagiarisms |
References added….. [Smulders and De Klerk 2011]. Smulders MJM, De Klerk GJ: Epigenetics in plant tissue culture. Plant Growth Regul. 2011, 63(2), 137-146. Necessary checks were made in terms of plagiarism with the “ITHENTICATE” programme. |
|
5. Authors: Tissue culture-induced variations include molecular and phenotypic changes induced in vitro due to continuous subculturing and stress derived from tissue cultures, which can induce epigenetic variations such as altered DNA methylation patterns.
Comment: Please, add references and check the text for plagiarisms |
References added and necessary checks were made in terms of plagiarism
Tissue culture-induced variations include molecular and phenotypic changes induced in vitro due to continuous subculturing and stress derived from tissue cultures, which can induce epigenetic variations such as altered DNA methylation patterns [Carrera et al., 2021]. |
|
6. Authors: Since DNA methylation plays an important role in gene expression and regulation of plant development, variants that arise during the tissue culture process due to inherited methylation changes may contribute to intra-specific phenotypic variation.
Comment: I understand this is your suggestion. Do you have any data that support the notion? |
Added an updated reference to support this sentence at the end of the sentence. [Shaikh et al., 2022]. Shaikh AA, Chachar S, Chachar M, Ahmed N, Guan C, Zhang P: Recent advances in DNA methylation and their potential breeding applications in plants. Horticulturae, 2022, 8(7), 562.
|
|
7. Authors: Various methods are available to detect changes in the genome-wide methylation pattern in tissue culture plants, most of which are based on extensive knowledge of an organism’s genome sequence [22]. However, methylation-sensitive amplification polymorphism (MSAP) – a modified amplified fragment length polymorphism (AFLP) technique does not require genome sequencing [23].
Comment: the text sound highly familiar. Have you checked it for plagiarisms? |
The sentence was checked for plagiarism and edited again. |
|
8. General. It would be great if you could separate Result and Discussion section into two independent part. In R section or maybe in methods graphical representation of your whole tissue culture and molecular experiment will be included. |
These sections have been separated from each other based on your suggestion. |
|
9. Comment: line 122. What does GTS mean?
|
Added meaning of GTS… the emergence of new bands, were calculated as the genomic template stability (GTS) value compared
|
|
10. Comment: line 129. 10-4 mM of estrogen hormone (52.2%). Should it be 10-4? Explain percentages in brackets, please? How did you get the number? Keep this in mind in the next sentences (130, 134, 135), too. |
Necessary checks and corrections were made by taking into account your suggestions throughout the entire text. |
|
11. Authors: There was a negative correlation between the percentage of GTS and MSH concentrations.
Comment: It is not apparent how quantification was conducted. Have you normalized your results somehow? Without it, it would be inadequate to make any comparisons (correlations). |
Dear reviewer; How the GTS value is calculated is presented in the method section and the evaluation is made accordingly. In addition, it was presented in Table 1 that the GTS value decreased due to increased hormone concentration. |
|
12. Authors: The administered MSH constructs caused changes in both the bean plant’s genetic structure and epigenetic profile. It is thought that this may be because MSHs affect plants up to inorganic content.
Comment: Could you be more precise, please? Do you mean metal ion cofactors? If so, please check my suggestion concerning antioxidants. |
We tried to add some explanatory reference sentences about this subject. However, inorganic elements have a vital role in the formation of organic substances. For example, the inorganic element P is a vital component of DNA and RNA [Erdal and Dumlupinar 2011].
In other study, it was reported that some mammalian sex hormones (MSHs) were successful in preventing genetic and epigenetic changes caused by certain chemicals in plants [37]. In addition, it is reported that MSHs stimulate antioxidant defense systems in plants [Turk 2021]. |
|
13. Author: For example, a study conducted on bean plants determined that mammalian sex hormones cause changes in the ratio of inorganic substances in the plant content [35]. Inorganic substances are essential structures that affect the functioning of enzymes.
Comment: Again, more info is needed on what you mean by INORGANIC SUBSTANCES. Please, be concrete. |
Dear reviewer; Added descriptive sentence…. The administered MSH constructs caused changes in both the bean plant’s genetic structure and epigenetic profile. It is thought that this may be because MSHs affect plants up to inorganic content. For example, a study conducted on bean plants determined that mammalian sex hormones cause changes in the ratio of inorganic substances in the plant content [35]. Inorganic substances are essential structures that affect the functioning of enzymes. Plants need inorganic elements (such as P, S, K, Fe and Ni) for synthesis reactions such as photosynthesis, protein synthesis and nucleic acid synthesis. It has been reported that there is an important relationship between inorganic elements and the speed of metabolic pathways in all living things. However, inorganic elements have a vital role in the formation of organic substances. For example, the inorganic element P is a vital component of DNA and RNA [Erdal and Dumlupinar 2011]. |
|
14. Comment: Figure 1. Would you give more details on control, please? How it is related to other samples? Have the other samples and control originated from the same donor plant? |
The explanations you want have been added to the text. ……….different concentrations (Control (without hormone), 10-4, 10-6 and 10-8 mM) of one of four mammalian sex hormones (estrogen, progesterone, 17 β-estradiol and testosterone). All samples, including the control group, originated from the same donor plant. |
|
15. Alternatively, you may consider population genetic approach. However, for that you should have quit a larch number of individuals for each experiment, check for PIC, I, He etc. Having stable parameters for control (via checking different sample sizes you may have a background for choosing an adequate sample size for your analysis). |
Dear reviewer; The primary aim of this study was to elucidate the effects of mammalian sex hormones (MSHs) on plant tissue culture, genomic instability, and DNA methylation of bean plants. Therefore, we did not analyze the parameters you mentioned. However, taking into account your very valuable suggestion, we will consider them in future studies. |

Reviewer 2 Report
The authors have tested the effects of mammalian sexual hormones on in vitro genetics and epigenetics of a bean plant. They describe as these hormones pal active roles in genomic stability and cause modifications.
The results are clear and I have no gig objections against the publication of the manuscript. Nevertheless, I would like that the authors address the following points.
It is unclear to this reviewer how is the occurrence of mammalian sexual hormones in plants. Do plants express genes encoding for these hormones? In this case they should play a physiological role. Or are the plants exposed to these hormones as consequence of anthropogenic activities? Please, clarify, this is important for readers non-specialized in plant physiology.
Are the authors using the term mammalian sex hormones when they really mean human sex hormones? Moreover, to my understanding estrogen is a general term for referring female sexual hormones and is not a specific chemical. Please, identify clearly the hormones stating in material and methods at least the following information: CAS number, purity of the preparations and provider.
I am not convinced about the suitability of the tittle because to my understanding DNA damage has not been tested in the manuscript. Such DNA damage is usually assessed through a Comet assay, and no results in this line were presented. Thus, I objet against the term DNA damage in tittle in other parts of the manuscript.
Any specific reason for the chosen of bean plant of model for this study? Please, clarify.
More information is needed in Tables 1 and 2. I understand that + means band present while – means band absent. I do not understand very well the concept of absent band. Absent as regard to which situation? Why this discrimination is not done in the control? In Table 2 I assume that H means HpaII and M means Mspl, please, stated in the legend.
Abbreviations. Please, pay more attention to the use of abbreviations. By example, mammalian sex hormones (MSH) is not defined in the abstract and it is used up to two times in the text before definition as abbreviation. In the middle, the term mammalian sex hormones (non-abbreviated) is use at least ones once the abbreviation had been used before. The term GTS is used by first time in line 122 and is defined in line 293.
Author Response
Responses to Comments of Reviewer 2
General Response:
Dear reviewer; We tried to respond to your valuable suggestions and comments in the best way possible. I hope it was a successful arrangement. The edits you want are highlighted in blue background color.
Sincerely
Dr. Aras Turkoglu, Dr. Peter Poczai et al.
|
Comments |
Responses: |
|
1. It is unclear to this reviewer how is the occurrence of mammalian sexual hormones in plants. Do plants express genes encoding for these hormones? In this case they should play a physiological role. Or are the plants exposed to these hormones as consequence of anthropogenic activities? Please, clarify, this is important for readers non-specialized in plant physiology. |
An explanatory sentence has been added to the introduction. “Mammalian sex hormones are released into the environment from natural and anthropogenic sources. In addition, these hormones can be treated as external application to plants [Upadhyay and Maier 2016].” |
|
2. Are the authors using the term mammalian sex hormones when they really mean human sex hormones? Moreover, to my understanding estrogen is a general term for referring female sexual hormones and is not a specific chemical. Please, identify clearly the hormones stating in material and methods at least the following information: CAS number, purity of the preparations and provider. |
Of course, the term mammalian sex hormones is used when they mean human sex hormones. In addition, although estrogen is a female hormone, estrogen, testosterone and 17β-estradiol, which are among the MCHs, are produced naturally in both plants and animals. Taking into account your suggestion, CAS numbers and other information regarding hormones have been added……[estrogen (C18H22O2: 270.37 g/mol; Sigma Aldrich; Product Number: E9750), progesterone (C21H30O2: 314.46 g/mol; Sigma Aldrich; Product Number: Y0001665) , 17 β-estradiol (C18H24O2: 272.38 g/mol; Sigma Aldrich; Product Number: E2758) and testosterone (C19H28O2: 288.42 g/mol; Sigma Aldrich; Product Number: Y0002163)]. |
|
3. I am not convinced about the suitability of the tittle because to my understanding DNA damage has not been tested in the manuscript. Such DNA damage is usually assessed through a Comet assay, and no results in this line were presented. Thus, I objet against the term DNA damage in tittle in other parts of the manuscript. |
The title has been revised based on your suggestion. New title: “The effect of mammalian sex hormones on polymorphism and genomic instability in the bean (Phaseolus vulgaris L.)”
|
|
4. Any specific reason for the chosen of bean plant of model for this study? Please, clarify. |
We chose this plant because of the lack of sufficient studies on this subject in beans and our interest in legumes. The reason for this was emphasized in the last paragraph of the introduction. “This study is the first to examine the epigenetic changes caused by MSHs in the bean genome with the iPBS and CRED-iPBS methods” |
|
5. More information is needed in Tables 1 and 2. I understand that + means band present while – means band absent. I do not understand very well the concept of absent band. Absent as regard to which situation? Why this discrimination is not done in the control? In Table 2 I assume that H means HpaII and M means Mspl, please, stated in the legend.
|
Added explanations of +/- at the end of Table 1.
An explanation has been added at the end of Table 2. *M- Msp I, H- Hpa II, *Control (without hormone). |
|
6. Abbreviations. Please, pay more attention to the use of abbreviations. By example, mammalian sex hormones (MSH) is not defined in the abstract and it is used up to two times in the text before definition as abbreviation. In the middle, the term mammalian sex hormones (non-abbreviated) is use at least ones once the abbreviation had been used before. The term GTS is used by first time in line 122 and is defined in line 293.
|
An explanation of the abbreviation has been added in the abstract section. “……..changes in DNA methylation and DNA stability in common bean exposed to mammalian sex hormones (MSH)”
Added abbreviation at first occurrence of GTS. This correction is highlighted in yellow because the first reviewer indicated it. ………emergence of new bands, were calculated as the genomic template stability (GTS) |

Round 2
Reviewer 2 Report
The authors have introduced changes in the manuscript that, together with clarification in their rebuttal letter, allow overcoming all the concerns that I raised during my first assessment.
One minor point, the authors state in the rebuttal letter that they added CAS number. I was unable to find such CAS number in the new version of the manuscript, but maybe this lack can be substituted with the references to catalog numbers of the providers.
Finally, as I comments in my first report, the term estrogen is a general term. The specific substance corresponding to the SIGMA-ALDRICH product number E9750 is estrone.